# Improving the Effective Coverage Space for Source-Free Domain Generalization via Visual-Language Models

## Abstract

With the widespread application of deep learning in computer vision, deep models often experience a significant drop in performance when facing unseen data, which negatively impacts their practical deployment. In this work, a dynamic feature construction and fusion method (DFCF) based on vision-language models is proposed for the task of source-free domain generalization. This method introduces the concept of Effective Coverage Space (ECS) and utilizes vision-language models to dynamically generate diverse feature representations and construct a virtual dataset, which transforms the source-free domain generalization into a supervised learning task. In the absence of source domain images, the effective coverage of the feature space is extended by improving the diversity of styles and features, thereby enhancing the model's adaptability to the unseen domain. Experimental results demonstrate that this method significantly improves performance of source-free domain generalization tasks across multiple datasets, effectively enhancing the generalization capability of the model.

## 1 Introduction

With the wide application of deep learning techniques in the field of computer vision, there exists a growing expectation for models to migrate to target domains and perform effectively without the need for fine-tuning. Consequently, the Domain Generalization (DG) technique (Blanchard et al., 2011) proposed to address this problem has emerged as a critical research direction aimed at enhancing model performance in unknown domains.

Notable progress has been achieved in domain generalization methods, such as those based on domain-invariant features (Li et al., 2018a;b; Ding et al., 2022; Liu et al., 2023; Xie et al., 2024), and data augmentation (Li & Spratling, 2022; Su et al., 2023; Xu et al., 2023; Ren et al., 2023a); however, several shortcomings remain. First, these methods typically depend on data from single or multiple source domains for training and impose stringent requirements on the data distribution and diversity of these domains. Consequently, effective generalization is often hindered when there is a substantial distribution difference between the target and source domains or when the source domain data lack sufficient diversity. Additionally, source-domain data may exhibit bias or noise, resulting in suboptimal performance of the trained model in the target domain. As a result, existing methods exhibit substantial limitations in addressing these challenges.

To address these problems, this paper introduces the concept of effective coverage space (ECS) to address the limitations associated with source domain generalization. The ECS refers to the area that encompasses correctly classifiable features within the feature space. The size of this area is closely linked to the model's generalization capability. Thus, a broader coverage area corresponds to enhanced generalization performance. Therefore, expanding the effective coverage of the feature space is essential for enhancing the model's adaptability in unknown domains.

Based on this concept, this paper proposes a dynamic method for feature construction and fusion (DFCF) within joint visual-language model. The overall structure of DFCF is illustrated in Fig. 1. Style information is dynamically generated using a Gaussian random generation method with multiple predefined text templates, and this generated style information is combined with category information to create diverse expressions for each style-category combination. Incorporating the concept

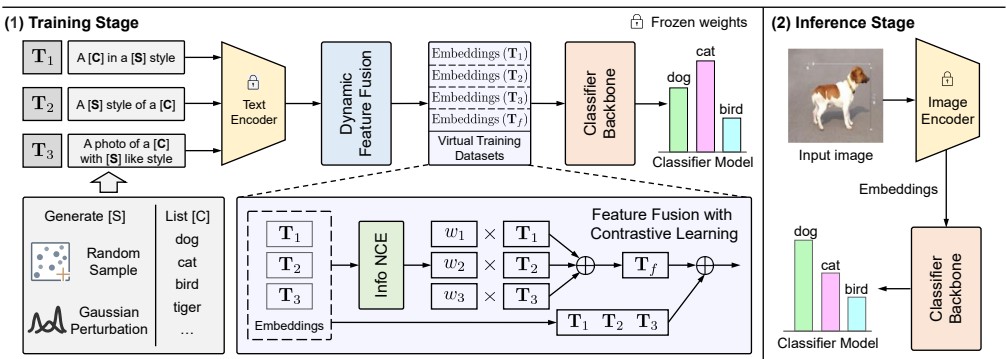

Figure 1: The overall diagram of the dynamic method for feature construction and fusion (DFCF).

of feature fusion, multiple expressions of the same style-category combination are fused to generate several vectors that contain style and category information. These vectors serve as feature representations and form a virtual dataset alongside the category labels provided during generation, facilitating the transformation of the source-free domain generalization problem into a supervised learning problem. In the absence of the source domain, the effective coverage of the feature space is expanded by enhancing the diversity of styles and features, thereby improving the model's adaptability and accuracy in the target domain. The main contributions are summarized as follows:

- The concept of ECS is introduced, and it is demonstrated both theoretically and experimentally that the model's generalization ability is closely linked to the size of its ECS. It is believed that expanding the size of ECS is crucial for enhancing the model's adaptability in unseen domains, thus providing a new perspective on the theoretical understanding of the domain generalization problem and the design of subsequent methods.

- The source-free domain generalization problem is transformed into a supervised learning problem via utilizing a joint vision-language model. Constructing a virtual dataset composed of text vectors and fusion vectors generated from multiple templates ensures that the model can be effectively trained even in the absence of source-domain data, demonstrating strong generalization ability and adaptability in the presence of unknown domains.

- A dynamic method for feature construction and fusion (DFCF) is proposed, which expands the ECS and enhances the model's adaptability across different domains by dynamically generating styles and fusing features from text vectors produced by multiple templates.

- Extensive experiments conducted on several standard datasets validate the superior performance of the proposed method in the source-free domain generalization task, particularly in the context of unseen domains, which exhibits substantial adaptability and robustness.

## 2 RELATED WORK

**Domain Generalization:** Domain Generalization (DG) is a core topic in the field of machine learning (Blanchard et al., 2011), with the aim of improving the performance of models in previously unseen domains. Traditional approaches rely on shared knowledge learned from source domain data to address tasks in the target domain. Researchers employ various techniques, including adversarial training (Li & Spratling, 2022; Ren et al., 2023a; Li et al., 2024), feature alignment (Chen et al., 2023a;b), and meta-learning (Qiao et al., 2020; Chen et al., 2023a), to minimize distributional discrepancies between the source and target domains. The goal of these approaches is to enable models to learn more robust and invariant feature representations, thereby improving their generalization ability across different data distributions. However, researchers are encountering growing challenges as data privacy concerns become more prominent and the cost of acquiring source domain data rises, prompting the exploration of source-free domain generalization methods.

**Source-Free Domain Generalization:** Source-free domain generalization is an emerging area within domain generalization techniques, aimed at enabling models to be trained without the use of source domain data for adaptation to unseen target domains (Niu et al., 2022). Current research emphasizes achieving this objective through the utilization of large pre-trained models, for instance,

by generating cross-domain consistent representations via visual-language modeling or employing prompt-driven approaches to produce diverse stylistic features that simulate distributional shifts (Cho et al., 2023). These studies not only circumvent the stringent data diversity and distribution requirements of traditional single or multi-domain generalization techniques but also offer novel perspectives for enhancing model performance in entirely new domains.

**Vision-Language Models:** In recent years, research on joint vision-language models (VLM) has attracted significant attention, through the use of contrastive learning, cross-modal attention mechanisms, and generative adversarial networks. These approaches combine image and language models to enhance the understanding of complex tasks. Models such as CLIP (Radford et al., 2021) and ERNIE-ViL (Yu et al., 2021) are widely employed in fields like image quizzing, image captioning, and multi-modal sentiment analysis. VLM have been applied in various applications, such as image classification and instance segmentation (Menon & Vondrick, 2023; Ren et al., 2023b). VLM integrate textual and visual information for source-free domain generalization, producing improved feature representations that enhance model adaptability.

## 3 METHODOLOGY

### 3.1 MOTIVATION

In domain generalization research, particularly in the context of source-free domain generalization, two core challenges arise: first, how to effectively model the diversity of unseen domains without access to source domain data; and second, how to construct a feature space with stronger generalization capability. These two issues are directly tied to the model's performance in unseen domains.

To address the first challenge, large-scale pre-trained models, such as BERT (Devlin et al., 2019) and CLIP (Radford et al., 2021), can be used as a foundation. These models have acquired a rich feature reserve through the input of large amounts of pre-training data. A method can be developed to filter potentially useful features from pre-trained feature spaces to construct a training set, thereby improving the model's generalization ability. In other words, source-free domain generalization does not imply the complete absence of data, but rather shifts the source of the training set from a directly provided dataset to one filtered from the feature space of pre-trained models. Therefore, once an effective method for filtering the feature space is identified, the problem of source-free domain generalization can be transformed into a straightforward supervised learning task, with the filtered features serving as the training set for classification model training.

To filter effective features, a method is required to identify them, leading to the introduction of the vision-language model. Since the VLM model has already achieved alignment between text and image features, different texts can be constructed from the desired categories and converted into aligned features via the VLM model, thereby covering the image features.

Based on the above ideas, the concept of ECS is proposed. This concept is used to mean the extent to which the model's feature space covers the unknown domain. In a 2D space, the region occupied by all feature points from each category is termed the ECS of the current category.

For the entire feature space $\boldsymbol{\Omega}$ of a pre-trained model, it is required to contain the feature representations of all categories. Assuming that the number of classifiable categories for the model is $K$, there should be $K$ sets of feature points $S_k$ $(1 \leq k \leq K)$ in $\boldsymbol{\Omega}$, each of which corresponds to a category. Let $n_k$ be the number of feature points contained in $S_k$, then $S_k$ can be simply represented as:

$$S_k = \{s_1, s_2, \ldots, s_{n_k}\}, \quad 1 \leq k \leq K, \tag{1}$$

where $s_i$ $(1 \leq i \leq n_k)$ stands for a specific feature point in $S_k$. Due to variations in the representation of features within the feature sets, intersections may occur between different feature sets. In an ideal scenario, a model performs better in recognition accuracy and generalization, intuitively implying two aspects: (1) *the features of each category should be as exhaustive as possible, so that the feature space is as fully populated by the features of all categories as possible*; and (2) *the features of different categories should minimize redundancy and maintain a certain distance to ensure that the descriptions of different categories remain relatively independent*, as shown in Fig. 2. Therefore, $S_k$ should satisfy the following constraint:

$$\bigcup_{k=1}^{K} S_k = \boldsymbol{\Omega} \quad \text{and} \quad \bigcap_{k=1}^{K} S_k = \varnothing. \tag{2}$$

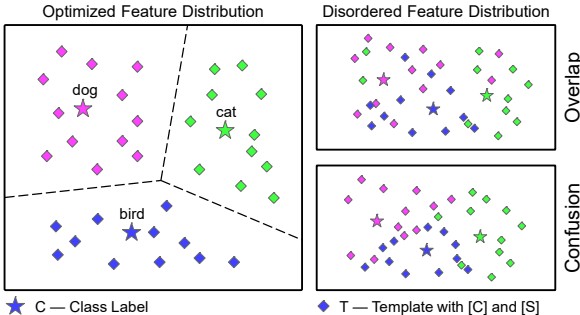

Figure 2: Optimized and disordered feature distribution.

Since the feature space contains categories that are much larger in number and scope than the number of categories that the model is required to classify, situations may arise in which the ECS cannot cover the entire feature space. However, for specific tasks, by expanding the size of the ECS for each category, it is expected that the model will be better adapted to unseen domains. From this perspective, the optimization problem of ECS can be formally formulated as the following:

$$\arg\max_{S_k} \left( \alpha \cdot \left| \bigcup_{k=1}^{K} S_k \right| - \beta \cdot \left| \bigcap_{k=1}^{K} S_k \right| \right), \tag{3}$$

where $\alpha$ and $\beta$ are tradeoff parameters.

As discussed earlier, the core goal of ECS optimization is to maximize the intra-class feature coverage (to adapt to unseen domains) and minimize inter-class feature overlap (to avoid confusion), which is formally described in Eq. 3. However, directly quantifying the total feature space $\Omega$ is infeasible due to its high dimensionality and abstract nature. To measure this, a computable proxy metric for ECS is proposed, which aligned with the aforementioned optimization goals and the contrastive learning paradigm. The ECS proxy metric $M$ is defined as:

$$M = \lambda_{intra} \frac{1}{K} \sum_{i=1}^{n_k} d(s_i, \mu_k) + \lambda_{inter} \sum_{1 \le i < j \le K} d(\mu_i, \mu_j) \tag{4}$$

where $\lambda_{intra}, \lambda_{inter} > 0$ are trade-off parameters for balancing the importance of intra-class and inter-class objectives. $d(a, b) = 1 - cos(a, b)$ denotes cosine distance selected for consistency with CLIP's normalized feature space to avoid bias from feature magnitude. $\mu_k = \frac{1}{n_k} \sum_{i=1}^{n_k} s_i$ denote the mean feature vector of $S_k$. The first term in Eq. 4, labeled as average intra-class coverage, is derived from the average distance between each feature point in $S_k$ and its centroid $\mu_k$. This term quantifies the coverage range of a category's features, where a larger value indicates more dispersed features and wider coverage of potential unseen domain variations. The second term in Eq. 4, referred to as total inter-class separation, equals the sum of pairwise distances between centroids of distinct categories. This term quantifies inter-class discriminability, where a larger value indicates smaller feature overlap and aligns with the ideal ECS property $\bigcap_{k=1}^{K} S_k = \varnothing$. From a physical perspective, an increase in $M$ implies broader intra-class coverage and better inter-class discriminability—exactly the core goal of expanding ECS.

### 3.2 STYLE GENERATION VIA STOCHASTIC PERTURBATION

To enhance feature coverage, we propose a novel style generation method via Gaussian stochastic perturbation to dynamically generate diverse style embeddings during model training. The method has two strategies: random and hybrid. When a style refresh is required, one strategy is selected randomly. If the random is chosen, a new set of style embeddings is generated, expanding the diversity of style representations beyond the base embeddings. If the hybrid is selected, a subset of embeddings is randomly chosen from the existing pool, which are then combined using linear weighting and injected with Gaussian noise to further increase style diversity and mitigate overfitting.

Specifically, let $E = \{e_1, e_2, \ldots, e_Q\}$ be the embedding pool of basic styles, from which $n$ embeddings are randomly selected. The generated style embedding, $\tilde{e}$, is given by:

$$\tilde{e} = \sum_{i \in \phi} w_i \cdot e_i + \epsilon, \epsilon \sim N(0, \sigma^2 I) \tag{5}$$

**Algorithm 1** Dynamic Style Generation

**Input**: The number of styles $n$, the pool of base style embeddings $E$

**Output**: List of generated styles $\tilde{E}$

1: **for** $k = 1$ to $n$ **do**
2:    choice = random(0,1)
3:    **if** choice $< 0.5$ **then**
4:      $\tilde{e} \leftarrow$ Perturb($E$)
5:       //Gaussian perturbation
6:    **else**
7:      $\tilde{e} \sim \mathcal{N}(0, I)$
8:       //Random Gaussian
9:    **end if**
10:   $\tilde{E} \leftarrow \tilde{E} \cup \tilde{e}$
11: **end for**
12: **return** $\tilde{E}$

**Algorithm 2** Contrastive Feature Fusion

**Input**: Templates $T$

**Output**: Dynamic weights $w$, fused feature $z_f$, fused loss $\mathcal{L}$

1: $w \leftarrow [0, 0, 0]$, $z_f \leftarrow 0$   //Initial
2: **while** not converged **do**
3:    **for** $i = 1$ to 3 **do**
4:      $z_i \leftarrow$ CLIP($T_i$)
5:      $\mathcal{L}_i \leftarrow \sum_{j \neq i}^{3} \text{Sim}(z_i, z_j)$
6:    **end for**
7:    $w_i \leftarrow \frac{\mathcal{L}_i}{\sum \mathcal{L}}$ //Dynamic weight calculation
8:    $z_f \leftarrow w_1 \cdot z_1 + w_2 \cdot z_2 + w_3 \cdot z_3$
9:    $\mathcal{L} \leftarrow w_1 \cdot \mathcal{L}_1 + w_2 \cdot \mathcal{L}_2 + w_3 \cdot \mathcal{L}_3$
10:   Update model parameters using $\mathcal{L}$
11: **end while**
12: **return** $w, z_f, \mathcal{L}$

where $w_i$ are random weights sampled and normalized from a uniform distribution, and $\epsilon$ denotes Gaussian noise. $Q = |E|$ is the sample size of $E$ and $\phi$ is the index set of $n$ selected embeddings.

Through the mechanism in Alg. 1, the dynamics and balance of style generation are achieved. On one hand, the stochastic mixing method generates diverse style representations by reorganizing the base style embeddings and injecting noise. On the other hand, the random initialization strategy expands the exploration space of style embeddings, thereby enhancing diversity for model training.

### 3.3 FEATURE FUSION VIA CONTRASTIVE LEARNING

Another aspect of enhancing feature coverage is the template that combines category and style. The template is processed by a text encoder that incorporates style (S) and category (C), resulting in feature representations containing comprehensive information. These representations are aggregated into clusters covering the feature space. Since different templates convey the same content using different forms, the features generated by each template will differ, leading to varying coverage of the feature space and thus impacting the final performance. Fig. 3 demonstrates this effect comparing images generated from different templates and style descriptions for the same category.

One way to improve feature coverage is to use more templates. However, if features from different templates are considered equally important, the model's representational capacity could be limited. Therefore, we propose a dynamically weighted feature fusion by contrastive learning. By dynamically adjusting the weights of each feature, the method can optimally utilize the unique information from each template to expand ECS more effectively and reasonably.

#### 3.3.1 SIMILARITY COMPUTATION BASED ON CONTRASTIVE LEARNING

As previously mentioned, the distribution of features $\{z_1, z_2, z_3\}$ generated by CLIP with different templates in the feature space may differ due to variations in expression across text templates. Evaluating the similarity between these distinct features is crucial for the effectiveness of the fused features. We quantify the similarity of features from different templates using contrastive learning loss, specifically the InfoNCE formulation (Oord et al., 2019).

$$\text{Sim}(z_i, z_j) = -\log \frac{\exp(\cos(z_i, z_j)/\tau)}{\sum_{k=1, k \neq i}^{3} \exp(\cos(z_i, z_k)/\tau)}. \tag{6}$$

InfoNCE uses noise-contrastive estimation to measure loss, maximizing positive and minimizing negative sample similarity. In the feature space, same-class samples cluster closely while different-class samples separate, aligning with ECS. $\cos(z_i, z_j)$ in Eq. 6 denotes the cosine similarity between the template features $z_i$ and $z_j$; $\tau$ is the temperature coefficient.

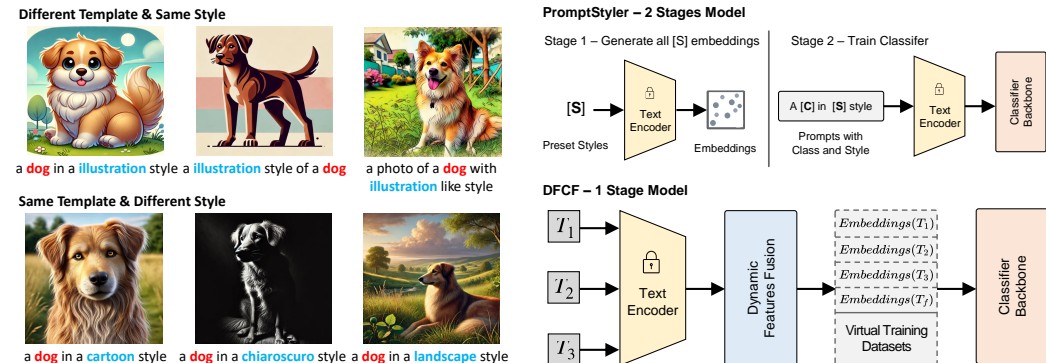

Figure 3: Effect: Templates & style descriptions.          Figure 4: Structure: PromptStyler & DFCF.

To synthesize a comprehensive measure of similarity between each template and the other templates, we calculate the total similarity score $\mathcal{L}_i = \sum_{\substack{j=1 \\ j \neq i}}^{3} \text{Sim}(z_i, z_j)$ for each text template. The total similarity score reflects the relative distribution of each template relative to other templates in the semantic space. A higher score indicates the template holds greater significance in the global space.

### 3.3.2 DYNAMIC WEIGHT COMPUTATION AND FUSION OF FEATURE WEIGHTS

After calculating the total similarity scores of the template features, we assign dynamic weights to each template to adaptively adjust its contribution to global feature fusion. Specifically, we normalize the total similarity scores to obtain the dynamic weights $w_i = \frac{\mathcal{L}_i}{\sum_{j=1}^{3} \mathcal{L}_j}$ for each template. The weights $w_i$ satisfy the condition $w_1 + w_2 + w_3 = 1$ with $w_i$ for all $i$. The final feature representation $z_f = w_1 \cdot z_1 + w_2 \cdot z_2 + w_3 \cdot z_3$ is obtained by performing a weighted fusion of all template features with dynamic weights. The fused loss $\mathcal{L} = w_1 \cdot \mathcal{L}_1 + w_2 \cdot \mathcal{L}_2 + w_3 \cdot \mathcal{L}_3$ is used for backward propagation and updating the model parameters to guide the model in its gradual optimization of key features. The pseudo-code in Alg. 2 illustrates the entire method.

## 4 EXPERIMENTS

### 4.1 EXPERIMENT SETS

#### 4.1.1 BENCHMARKS

We evaluate our approach using four widely recognized public datasets for domain generalization: PACS (Li et al., 2017), VLCS (Choi et al., 2022), OfficeHome (Venkateswara et al., 2017), and DomainNet (Leventidis et al., 2023). These datasets represent a gradient across size, class distribution, domain variation, and complexity, effectively addressing both the fundamental generalization problem and model performance in more challenging scenarios. Arranged from PACS to DomainNet, they encompass not only simple visual style variations but also extensive cross-domain tasks, reflecting a gradual transition from experimental validation to real-world challenges. This framework facilitates a comprehensive analysis of models' strengths and weaknesses across scenarios.

#### 4.1.2 BASELINE

We select PromptStyler (Cho et al., 2023) as the baseline for source-free domain generalization and evaluate its performance across different backbones. PromptStyler is a representative approach in domain generalization research, simulating domain diversity via style prompts to enhance adaptability to unknown target domains as shown in Fig. 4. Since PromptStyler's code is unavailable, we reproduce its results. For comprehensive comparison, we include other mainstream methods, such as MLDG (Min et al., 2022), SagNet (Nam et al., 2021), SelfReg (Kim et al., 2021), MIRO (Cha et al., 2022), Text Regularization (TR)(Zhang et al., 2024), Model Ratatouille (MR)(Rame et al., 2023), ZS-CLIP (Radford et al., 2021), CAD (Dubois et al., 2021), DCLIP (Menon & Vondrick, 2023), Cp-CLIP (Ren et al., 2023b), DPStyler(Tang et al., 2025) to build a multi-dimensional benchmark.

### 4.1.3 SETUP

We adopt a dynamic learning rate strategy balancing rapid initial convergence and late-stage optimization stability, enhancing model generalization via progress-based adaptation. Three mainstream models of varying sizes serve as classification backbones: ResNet-50 (He et al., 2016), ViT-B/16, and ViT-L/14 (Dosovitskiy et al., 2020). All models are initialized with pre-trained weights provided by CLIP, which were trained using large-scale image-text alignment, thereby equipping them with stronger semantic comprehension, cross-domain adaptability, and a foundational feature space. Experiments were conducted on an NVIDIA RTX 3090 GPU and an Intel Xeon Platinum 8255C CPU. These components meet the computational requirements of large-scale data processing and model optimization, ensuring robust hardware support for reliable experimental results.

## 4.2 EVALUATIONS

### 4.2.1 MAIN RESULTS

As shown in Table 1, we perform three average tests on the benchmarks. On ResNet-50, the average accuracy of the PromptStyler method is 73.3%, while our method achieves 73.8%, resulting in an improvement of 0.5%. Specifically, PACS and VLCS show improvements of 0.4% and 1.4%, respectively, demonstrating the effectiveness of our method in addressing style variations and multi-source data. The overall performance remains stable, with slightly fluctuating results observed in OfficeHome and DomainNet. Although the results on DomainNet exhibit minor fluctuations, overall performance stability is maintained. ResNet-50, as a traditional convolutional neural network, has a local receptive field that restricts its ability to model global context, particularly in complex domain distributions such as DomainNet. Consequently, performance gains are more moderate under this architecture, indicating that our method enhances generalization capabilities based on existing feature representations, albeit limited by the model's expressive power.

When upgrading to ViT-B/16 (base-scale), the PromptStyler method achieves an average accuracy of 79.8%, while our method reaches 80.3%, showing a 0.5% improvement. Notably, the improvements of 0.8% and 0.9% on the OfficeHome and DomainNet datasets, respectively, with stable performance on PACS and VLCS, indicate that our method better adapts to domain distribution differences and leverages stronger global modeling capabilities. Moreover, it provides enhanced feature information for the backbone network, thus further boosting cross-domain generalization performance.

Under the ViT-L/14 (large-scale) architecture, the average accuracy of the PromptStyler method is 82.2%, whereas our method achieves 83.2%, representing a significant 1.0% improvement. The notable increases of 1.8% and 1.7% on VLCS and OfficeHome, respectively, along with stable performance growth in the DomainNet dataset, fully demonstrate the strong potential of our method under the robust modeling capabilities of large-scale backbone networks. This reflects our method's generalization advantage for complex cross-domain tasks.

From ResNet-50 to ViT-L/14, the increase in model complexity boosts overall performance, indicating that stronger global modeling capabilities, combined with richer feature representations, are crucial for domain generalization tasks. Our approach outperforms the PromptStyler method across all models, achieving the largest performance gain specifically on ViT-L/14. For PACS, which exhibits large stylistic differences, and VLCS, which displays significant source differences, our method maintains consistent performance across architectures. In contrast, for OfficeHome and DomainNet, which involve numerous categories and complex domains, the combination of ViT-L/14 with our method reveals more pronounced performance advantages, indicating that our method is better suited for high-dimensional distributions and complex cross-domain tasks. As shown in Appendix, detailed results for each domain in PACS, VLCS, OfficeHome, and DomainNet are presented to further evaluate the effectiveness of DFCF. The letters in the table titles are abbreviations of the data domain names, with the full names provided below each table.

Through the experiments and analysis, our method demonstrates superior performance across various architectures, with performance gains becoming more evident as model complexity increases. This indicates that our method effectively utilizes pre-trained features with global modeling capabilities, offering an efficient and scalable solution for source-free domain generalization tasks.

Table 1: Performance comparison of different methods on various datasets.

| Backbone | Methods | Accuracy (%) | | | | Average Accuracy | Improvement (%) |
|---|---|---|---|---|---|---|---|
| | | PACS | VLCS | OfficeHome | DomainNet | | |
| ResNet50 (ImageNet) | MLDG | 84.9 | 77.2 | 66.8 | 41.2 | 67.5 | 0 |
| | SagNet | 86.3 | 77.8 | 68.1 | 40.3 | 68.1 | +0.6 |
| | SelfReg | 85.6 | 77.8 | 67.9 | 42.8 | 68.5 | +1.0 |
| | MIRO | 85.4 | 79.0 | 70.5 | 44.3 | 69.8 | +2.3 |
| | TR | 87.2 | 80.3 | 70.4 | 44.0 | 70.5 | +3.0 |
| | MR | 89.8 | 78.3 | 73.5 | 47.7 | 72.3 | +4.8 |
| ResNet50 (CLIP) | ZS-CLIP(C) | 90.6 | 79.4 | 67.4 | 45.9 | 70.8 | 0 |
| | CAD | 90.0 | 81.2 | 70.5 | 45.5 | 71.8 | +1.0 |
| | ZS-CLIP(PC) | 90.7 | 82.0 | 71.1 | 46.1 | 72.5 | +1.7 |
| | TR | 91.3 | 82.8 | 71.6 | 44.6 | 72.6 | +1.8 |
| | DPStyler | 92.1 | 81.3 | 70.5 | 47.6 | 72.9 | +2.1 |
| | PromptStyler | 93.1 | 82.2 | 71.0 | 46.9 | 73.3 | +2.5 |
| | **DFCF (OURS)** | **93.5** | **83.6** | **72.2** | 46.0 | **73.8** | **+3.0** |
| ViT-B/16 (CLIP) | ZS-CLIP(C) | 95.6 | 76.2 | 79.6 | 57.4 | 77.2 | 0 |
| | DCLIP | 91.4 | 81.3 | 81.9 | 56.6 | 77.8 | +0.6 |
| | MIRO | 95.6 | 82.2 | 82.5 | 54.0 | 78.6 | +1.4 |
| | Cp-CLIP | 96.5 | 82.7 | 79.9 | 57.2 | 79.1 | +1.9 |
| | ZS-CLIP(PC) | 96.0 | 83.0 | 81.8 | 57.2 | 79.5 | +2.3 |
| | DPStyler | 96.6 | 81.1 | 81.6 | 58.8 | 79.5 | +2.3 |
| | TR | 95.9 | 83.0 | 82.3 | **57.9** | 79.8 | +2.6 |
| | PromptStyler | 96.8 | 83.7 | 81.8 | 56.7 | 79.8 | +2.6 |
| | **DFCF (OURS)** | **97.0** | **84.0** | **82.6** | 57.6 | **80.3** | **+3.1** |
| ViT-L/14 (CLIP) | ZS-CLIP(C) | 97.6 | 77.5 | 85.7 | 63.1 | 81.0 | 0 |
| | ZS-CLIP(PC) | 98.3 | 81.9 | 86.6 | 63.0 | 82.5 | +1.5 |
| | PromptStyler | 98.4 | 81.3 | 86.4 | 62.9 | 82.2 | +1.2 |
| | DPStyler | 98.4 | 81.6 | 87.5 | 64.1 | 82.9 | +1.9 |
| | **DFCF (OURS)** | **98.5** | **83.1** | **88.1** | 63.2 | **83.2** | **+2.2** |

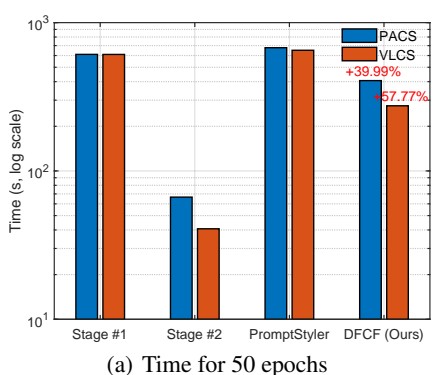

(a) Time for 50 epochs

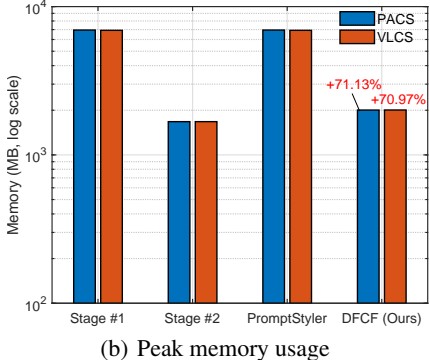

(b) Peak memory usage

Figure 5: Model efficiency comparison on PACS & VLCS. Stage #1 & #2 are PromptStyler phases.

### 4.2.2 COMPUTATIONAL RESULTS

In terms of training time and maximum memory usage, both PromptStyler and our method utilize the same batch size of 256. Fig. 5 shows that for 50 epochs on the VLCS dataset, our method achieves a time reduction of up to 57.7% and a decrease in memory usage of over 70%, demonstrating efficiency. Additionally, since PromptStyler employs a two-stage approach, the second stage operates on the results from the first stage. Thus, while the second stage may reach convergence quickly, the overall training time is extended due to the first stage needing to be completed. In contrast, our method follows a one-stage approach that integrates the dynamic style generation, feature fusion, and backbone training into a single process. This allows for rapid completion of training once the backbone network converges, resulting in a reduction in overall training time.

### 4.3 ABLATION STUDY

In the ablation experiments, we thoroughly evaluate the impact of three modules: dynamic styles, multi-templates, and feature fusion on model performance. Table 2 indicates that the introduction of dynamic styles improves the average accuracy from 73.3% in the baseline to 73.7%. This increase is particularly notable on the VLCS dataset, suggesting that dynamic styles effectively extend the model's coverage space by adapting to style variations across different domains. Similarly, the in-

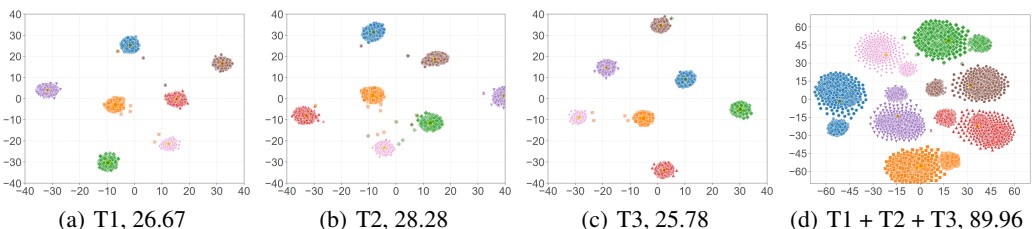

(a) T1, 26.67    (b) T2, 28.28    (c) T3, 25.78    (d) T1 + T2 + T3, 89.96

Figure 6: The feature distributions using one single template (a) ∼ (c), and that using the combination of all templates (d). The number in caption is proxy metric $M$.

corporation of diverse templates raises the average accuracy to 74.0%, achieving 93.5% and 83.2% on the PACS and VLCS datasets, respectively. This improvement implies that enriching the diversity of templates enables the model to learn more generalized and domain-invariant features. When dynamic styles are used with diverse templates, the performance remains stable at 74.0%, demonstrating the synergistic effect of these two components; dynamic styles broaden the adaptation range for style variations, while diverse templates further enhance the richness of data representation.

Fig. 6 illustrates the distribution of feature points under different conditions. The t-SNE method is employed to visualize the vectors input to the backbone network classifier, downscaled and color-coded for clarity. The images clearly show that the ECS incorporating multiple templates and feature fusion is considerably larger than the ECS in the single-template case. To quantitatively verify this observation, the $M$ defined in Eq. 4 is calculated for different template configurations based on the feature data in Fig. 6. The results show that: single template T1 yields 26.67, T2 yields 28.28, T3 yields 25.78, while the combined configuration of all templates achieves 89.96. This numerical trend aligns with the visual pattern in Fig. 6, confirming that integrating multiple templates significantly expands the effective coverage space.

Table 2: Performance comparison of ablation study.

| D | M | F | PACS | VLCS | OfficeHome | DomainNet | Avg. |
|---|---|---|------|------|------------|-----------|------|
|   |   |   | 93.1 | 82.2 | 71.0 | 46.9 | 73.3 |
| ✓ |   |   | 93.2 | 83.3 | 70.8 | 47.3 | 73.7 |
|   | ✓ |   | 93.5 | 83.2 | 72.1 | 47.1 | 74.0 |
| ✓ | ✓ |   | 93.5 | 83.5 | 72.1 | 47.0 | 74.0 |
|   | ✓ | ✓ | 93.5 | 83.0 | 72.1 | 46.0 | 73.7 |
| ✓ | ✓ | ✓ | 93.5 | 83.6 | 72.3 | 46.0 | 73.9 |

**D** - Dynamic Style    **M** - Multiple Template    **F** - Features Fusion

However, following the introduction of the feature fusion module, average performance does not improve and even decreases on the large-scale DomainNet dataset (from 47.1% to 46.0%). This phenomenon can be attributed to the introduction of redundant information in complex distribution scenarios, which leads to an overly dispersed feature distribution that narrows the ECS and diminishes generalization ability. Nevertheless, we retain this module due to its stable performance on small-scale datasets (like PACS and VLCS), its capacity to enhance the richness of feature representations, and its role in extending the upper limit of the coverage space in conjunction with dynamic styles and diverse templates. This ensures balanced and robust performance in general.

To validate the effectiveness of the feature generation algorithm, we conduct an experimental comparison evaluating the impact of using the feature generation strategy. Appendix demonstrates that combining the flexibility of random generation with the benefits of Gaussian perturbation significantly improves the model performance. Specifically, the average accuracy of four datasets increased by 1.3%, confirming the effectiveness of the proposed method.

## 5 CONCLUSION

In this work, we propose a novel dynamic feature construction and fusion framework for source-free domain generalization in vision-language models. We introduce the concept of ECS, a principled approach that establishes a critical link between the generalizability of the model and the coverage of the feature space. By dynamically synthesizing diverse feature representations and constructing virtual training datasets, our method effectively reformulates source-free domain generalization as a supervised learning problem. Extensive experiments demonstrate that this approach substantially improves the model's adaptability to unseen domains, as verified by the results.

In future work, we will focus on optimizing the feature fusion strategy to minimize irrelevant information and enhance model accuracy. We will also incorporate quantitative ECS analysis to enhance the model's generalization performance in complex cross-domain tasks.

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

## A    USE OF LLMs

We declare that Large Language Models (Doubao and Gemini) were used in the preparation of this manuscript, with their application strictly limited to text polishing and grammatical error checking. The core research content—including ideas, experimental design, data analysis, and conclusion derivation—is the authors' original work; LLMs only optimized written expression (e.g., refining sentence structure, aligning with academic conventions) without participating in substantive research or content creation. All LLM-assisted revisions were rigorously reviewed by the authors to ensure accuracy and compliance with academic norms.

## B    THE DETAILS OF DATASETS IN EXPERIMENTS

Table 3 shows the specific information of each dataset used in the experiment, including the number of data domains, categories, and data volume included.

Table 3: The details of datasets used in our experiments, including number of domains, classes, and image counts.

| dataset | domains | classes | image numbers |
|---------|---------|---------|---------------|
| PACS | 4 | 7 | 9,991 |
| VLCS | 4 | 5 | 10,729 |
| OfficeHome | 4 | 65 | 15,588 |
| DomainNet | 6 | 345 | 586,575 |

## C    COMPARISON OF RANDOM AND HYBRID STRATEGIES

Table 4 shows the comparison of random and hybird strategies in DFCF.

Table 4: Comparison of random and hybrid strategies.

| Method | PACS | VLCS | Office home | Domain Net | Avg. |
|--------|------|------|-------------|------------|------|
| hybrid | 92.5 | 83.0 | 72.0 | 46.0 | 72.6 |
| random + hybrid | 92.5 | 83.6 | 72.3 | 46.0 | 73.9 |

## D    DOMAIN PERFORMANCE COMPARISON OF DIFFERENT METHODS ON PACS, VLCS, OFFICEHOME AND DOMAINNET.

Table 5-8 show the domain performance comparison of different methods on PACS, VLCS, Office-Home and DomainNet.

Table 5: Domain performance comparison of different methods on PACS.

| Backbone | Methods | Accuracy (%) | | | | Average Accuracy |
|---|---|---|---|---|---|---|
| | | A | C | P | S | |
| ResNet50 (ImageNet) | SelfReg | 87.9 | 79.4 | 96.8 | 78.3 | 85.6 |
| | Model Rattatouille | 90.6 | 84.7 | 98.8 | 85.0 | 89.8 |
| ResNet50 (CLIP) | ZS-CLIP(C) | 88.9 | 94.4 | 99.3 | 79.8 | 90.6 |
| | ZS-CLIP(PC) | 90.8 | 93.3 | 99.4 | 79.3 | 90.7 |
| | PromptStyler | 93.6 | **95.2** | 99.3 | 84.2 | 93.1 |
| | **DFCF (OURS)** | **93.9** | 94.5 | **99.5** | **86.0** | **93.5** |
| ViT-B/16 (CLIP) | ZS-CLIP(C) | 96.4 | 98.7 | **99.9** | 87.5 | 95.6 |
| | ZS-CLIP(PC) | 97.3 | 99.0 | **99.9** | 88.0 | 96.1 |
| | PromptStyler | 97.4 | **99.1** | **99.9** | 90.7 | 96.8 |
| | **DFCF (OURS)** | **97.5** | 99.0 | **99.9** | **91.4** | **97.0** |
| ViT-L/14 (CLIP) | ZS-CLIP(C) | 97.2 | 99.4 | 99.9 | 93.9 | 97.6 |
| | ZS-CLIP(PC) | 98.6 | 99.5 | 99.9 | 95.3 | 98.3 |
| | PromptStyler | **98.8** | **99.8** | **100.0** | 95.1 | 98.4 |
| | **DFCF (OURS)** | **98.8** | 99.7 | **100.0** | **95.5** | **98.5** |

**A** - Art       **C** - Cartoon       **P** - Photo       **S** - Sketch

Table 6: Domain performance comparison of different methods on VLCS.

| Backbone | Methods | Accuracy (%) | | | | Average Accuracy |
|---|---|---|---|---|---|---|
| | | C | L | S | V | |
| ResNet50 (ImageNet) | SelfReg | 96.7 | 65.2 | 73.1 | 76.2 | 77.8 |
| | Model Rattatouille | 99.3 | 60.4 | 73.9 | 79.5 | 78.3 |
| ResNet50 (CLIP) | ZS-CLIP(C) | 99.5 | 67.8 | 69.5 | 80.8 | 79.4 |
| | ZS-CLIP(PC) | 99.8 | 69.6 | 71.0 | 87.7 | 82.0 |
| | PromptStyler | **100.0** | **72.5** | 67.9 | 88.4 | 82.2 |
| | **DFCF (OURS)** | **100.0** | 72.1 | **72.5** | **89.7** | **83.6** |
| ViT-B/16 (CLIP) | ZS-CLIP(C) | 99.8 | 60.9 | 69.8 | 74.1 | 76.2 |
| | ZS-CLIP(PC) | **100.0** | 70.0 | 74.1 | 88.0 | 83.0 |
| | PromptStyler | **100.0** | **72.5** | 72.4 | **89.9** | 83.7 |
| | **DFCF (OURS)** | **100.0** | 69.0 | **77.0** | 89.8 | **84.0** |
| ViT-L/14 (CLIP) | ZS-CLIP(C) | **100.0** | 57.5 | 70.5 | 82.1 | 77.5 |
| | ZS-CLIP(PC) | **100.0** | 70.8 | 68.6 | **88.1** | 81.9 |
| | PromptStyler | **100.0** | 66.3 | 71.8 | 87.0 | 81.3 |
| | **DFCF (OURS)** | **100.0** | 69.6 | **75.3** | 87.6 | **83.1** |

**C** - Caltech       **L** - Labelme       **S** - SUN09       **V** - VOC2007

Table 7: Domain performance comparison of different methods on OfficeHome.

| Backbone | Methods | Accuracy (%) | | | | Average Accuracy |
|---|---|---|---|---|---|---|
| | | A | C | P | R | |
| ResNet50 (ImageNet) | SelfReg | 63.6 | 53.1 | 76.9 | 78.1 | 67.9 |
| ResNet50 (CLIP) | ZS-CLIP(C) | 68.7 | 44.4 | 77.1 | 79.5 | 67.4 |
| | ZS-CLIP(PC) | 71.1 | 50.0 | 81.3 | 82.0 | 71.1 |
| | PromptStyler | 71.3 | 48.8 | **81.9** | 82.2 | 71.1 |
| | **DFCF (OURS)** | **73.4** | **50.4** | 81.8 | **83.3** | **72.2** |
| ViT-B/16 (CLIP) | ZS-CLIP(C) | 80.9 | 64.3 | 85.9 | 87.2 | 79.6 |
| | ZS-CLIP(PC) | 83.1 | 65.8 | 89.1 | 89.2 | 81.8 |
| | PromptStyler | 81.8 | 66.0 | 89.7 | 89.6 | 81.8 |
| | **DFCF (OURS)** | **83.6** | **67.2** | **89.8** | **89.9** | **82.6** |
| ViT-L/14 (CLIP) | ZS-CLIP(C) | 86.4 | 72.3 | 92.3 | 91.8 | 85.7 |
| | ZS-CLIP(PC) | 86.8 | 73.6 | 92.9 | **93.2** | 86.6 |
| | PromptStyler | 86.3 | 73.5 | 93.5 | 92.2 | 86.4 |
| | **DFCF (OURS)** | **87.6** | **77.8** | **93.9** | 93.0 | **88.1** |

**A** - Art       **C** - Clipart       **P** - Product       **R** - Real

Table 8: Domain performance comparison of different methods on DomainNet.

| Backbone | Methods | Accuracy (%) | | | | | | Average Accuracy |
|---|---|---|---|---|---|---|---|---|
| | | C | I | P | Q | R | S | |
| **ResNet50 (ImageNet)** | SelfReg | 60.7 | 21.6 | 49.4 | 12.7 | 60.7 | 51.7 | 42.8 |
| | Model Rattatouille | 66.1 | 23.1 | 55.5 | 16.7 | 68.5 | 56.0 | 47.7 |
| **ResNet50 (CLIP)** | ZS-CLIP(C) | 52.8 | 40.1 | 52.9 | **6.5** | 75.3 | 47.6 | 45.9 |
| | ZS-CLIP(PC) | 53.1 | 39.6 | 52.7 | 5.6 | **76.8** | 48.5 | 46.1 |
| | PromptStyler | **53.9** | **41.4** | **54.6** | 5.6 | **76.8** | **49.2** | **46.9** |
| | **DFCF (OURS)** | 52.0 | 41.0 | 53.3 | 6.3 | 75.1 | 48.0 | 46.0 |
| **ViT-B/16 (CLIP)** | ZS-CLIP(C) | 70.2 | 48.9 | **65.7** | **14.3** | 82.4 | 62.7 | 57.4 |
| | ZS-CLIP(PC) | **70.4** | 47.3 | 65.0 | 13.5 | **83.3** | **63.6** | 57.2 |
| | PromptStyler | 70.1 | 47.7 | 65.1 | 12.5 | 82.3 | 62.3 | 56.7 |
| | **DFCF (OURS)** | 70.1 | **50.0** | 65.5 | 14.2 | 82.6 | 63.2 | **57.6** |
| **ViT-L/14 (CLIP)** | ZS-CLIP(C) | 77.6 | 52.7 | **71.0** | 21.6 | 58.9 | 70.0 | 63.1 |
| | ZS-CLIP(PC) | **78.3** | 50.6 | 69.0 | **22.4** | **86.3** | **71.5** | 63.0 |
| | PromptStyler | 77.5 | 52.3 | 70.8 | 21.0 | 86.1 | 69.5 | 62.9 |
| | **DFCF (OURS)** | 77.5 | **53.8** | **71.0** | 20.9 | 85.9 | 70.2 | **63.2** |

**C** - Clipart    **I** - Infograph    **P** - Painting    **Q** - Qucikdraw
**R** - Real    **S** - Sketch

