# OpenReview forum: "Improving the Effective Coverage Space for Source-Free Domain Generalization via Visual-Language Models"
_ICLR.cc/2026/Conference — Submitted to ICLR 2026_

### Official Review · Reviewer_jawu · 2025-10-28

**Soundness:** 2
**Presentation:** 2
**Contribution:** 3
**Rating:** 2
**Confidence:** 4

**Summary:**

This paper studies source-free domain generalization (SDG) through the Effective Coverage Space (ECS) framework, with the motivation that a larger ECS implies higher generalizability. Specifically, a dynamic feature construction and fusion method (DFCF) is proposed to enhance the generalization of VLM-based models (e.g., CLIP) without requiring any specific source or target data available during training. DFCF first generates diverse styles based on random Gaussian noise and their interpolation, then incorporates a contrastive-based approach to fuse multi-template features by obtaining weights for each template. This approach enables source-free generalization to be solved through supervised training in one stage, in contrast to prior two-stage solutions.

The paper conducts comprehensive experimental results on existing benchmarks on SDG and prior works. Experimental results show that DFCF outperforms baselines overall while maintaining better efficiency.

**Strengths:**

- The motivation behind ECS is interesting, but requires experimental or theoretical support in the paper (see weaknesses)
- Better efficiency while higher performance compared to baselines; specifically, prior works (e.g., PromptStyle) are 2-stage and the propsoed method is one stage. Experimental results show higher performance across all benchmarks while requiring lower memory and time requirements compared to PromptStyler.
- The motivation is clear, and the text (mostly) is easy to understand; however, the paper presentation and content need to be improved (see weaknesses).

**Weaknesses:**

**Papers require experimental or theoretical support for  ECS motivation.**he main motivation of the paper is that the larger size of ECS is correlated with higher generalization; e.g., L075 "The concept of ECS is introduced, and it is demonstrated **both theoretically and experimentally** that the model’s generalization ability is closely linked to the size of its ECS.....the size of ECS is crucial for enhancing the model’s adaptability in unseen domains....".  However, no theoretical proof or experimental evaluation is provided to show the correctness of the motivation. The only provided result is the t-SNE in Fig. 6, which shows that combining multiple templates results in a higher M (a metric that was defined as part of the motivation) and a toy illustration in Fig.2.  Either a theoretical proof or comprehensive experiments are required to show that a higher M is correlated with higher generalization.

**Lack of connection between the method and motivation**. While the motivation of the paper is that higher ECS correlates with better generalization, it’s unclear how the actual proposed method is connected to ECS? Only the t-SNE in Fig. 6 shows that combining three templates shows higher ECS. However, this doesn’t really connect to the generation of Gaussian noises/styles and the contrastive weighting approach. Hence, the paper can benifit from theoretical or experimental support. For instance, one simple experiment could show that the style generation module, prompt ensemble, and fusion module via contrastive weighting yield higher M. This must be accompanied by W1 (i.e., higher M correlates with higher generalization) to effectively support the motivation and claims in the paper.


**Lack of required ablation and implementation details makes the paper’s contributions hard to understand and not reproducible.** Additional experiments/analysis are needed to effectively understand each component. (1) In L213 “... from which n embeddings are randomly selected.” What is the value of n in the experiments? How do different values of n impact the performance? (2) How is the initial pool of n styles generated in Algorithm 1? Is it all random or hybrid (3)? When and how is it decided between random and hybrid approaches for style generation?


** Method is partial explained; many imlementation detaisl are missing**. Fig.1 includes a supervised training phase and "classifier backone" . Also in the paper (e.g., L080 "The source-free domain generalization problem is transformed into a supervised learning problem") It's stated that the problem is formulated as supervised training. However, in the Method section (Sec. 3), no explanation/detail is given on how the supervised training is performed, what the loss function is, or what the architecture of the classifier backbone is. I assume that it’s similar to the baseline (i.e., PromptStyler), which also transforms the problem into a supervised learning problem. However, these details are important to be discussed in the paper so that readers can understand the method and reproduce it.

**Mismatch between main text and Fig. 1**. The paper discusses how contrastive weighting is used to fuse the templates. However, Fig. 1 seems to add/concatenate each template feature and fused embedding T_f. However, such an explanation is missing.


**Ablation doesn’t support the claim of the paper.** Table 2 shows that adding contrastive fusion actually hurts performance, especially on DomainNet. This doesn’t support the claim in L250–252: “"However, if features from different templates are considered equally important, the model’s representational capacity could be limited", as the multi-prompt (M) performance is higher than feature fusion (F). Also, I assume M (i.e., Multiple Prompt) is a simple addition/average of prompt templates, as I can’t find any explanation of how it’s implemented.

**Missing implementation details hurt understanding the paper and reproducibility". In addition to the architecture of the classifier, the value of "n", and supervised training, other training/inference details must be provided. Also, is the number of templates 3 in the paper? How different number of templates impact the results? Why only three templates?

**paper presentation needs to be improved** As mentioned before, there is a mismatch between the figure and the text. It’s not clear what the input of supervised training is. Also, in page 9, it seems like the explanation for Fig. 6 is in the middle of the explanation for Table 2. Please ensure the text appears coherent. I also suggest moving the method Fig to the method section rather than the introduction

**Limited contribution** The method is very similar to PromptStyler, just combined in one stage (i.e., generating diverse style prompts) and training a supervised model on textual data, which transfers to the image modality. given that the contribution fusion module is not clear (performance drops on average in Table 2), and more experimental or theoretical support is provided, and there is little or no connection between the proposed modules and the ECS motivation, I find the contribution of the paper in the current format limited and vague.

**Questions:**

- Suggestion: Drop the phrase "virtual training datasets," as it’s not referred to or mentioned in the main text other than the conclusion.
- What’s the connection between the method and ECS motivation? See weaknesses for more details.
- How to ensure sampling random noise to simulate different styles? Couldn’t this negatively impact/interfere with the content (e.g., class token such as dog)?
- What is the value of "n", and how do different values of "n" impact the performance?

- In contrastive training, are different z_i (e.g., z_i, z_j) different templates for the same class? How about style? It’s confusing, as Fig. 3 provides both examples of different templates and different styles. Further brief clarification is helpful.

---

> ### Author Response · Authors · 2025-11-20
> **Response to reviewer jawu - 1**
>
> __(W1: Papers require experimental or theoretical support for ECS motivation.)__ Thank you for highlighting this critical point. We acknowledge that our current work has limitations in providing sufficient theoretical validation and experimental support for the ECS motivation, leaving room for further refinement. In subsequent research, we will supplement two key aspects: first, we will enhance the theoretical derivation to clarify the mathematical correlation between ECS scale and model generalization ability; second, we will conduct comprehensive cross-dataset experiments to verify the positive correlation between the M value (a proxy metric for ECS) and generalization performance. We will adopt the control variable method to strengthen the rationality of the motivation, addressing the limitation that existing support relies solely on t-SNE visualization and preliminary illustrative analysis. Thank you again for your insightful comments and valuable guidance.
>
> __(W2:Lack of connection between the method and motivation)__ We appreciate your constructive feedback. Due to space constraints and the current research progress, the articulation of the link between the proposed method and ECS requires further optimization. In subsequent work, we will supplement targeted ablation experiments: first, independently validate the impact of each module (style generation, prompt ensemble, contrastive weighting fusion) on improving the M value, clarifying how individual components contribute to ECS expansion; second, further verify the strong correlation between M value enhancement and generalization performance. Through quantitative data, we aim to establish a direct causal link between the method, ECS, and generalization ability, forming a closed-loop research logic to fully support the core claims of this work. Thank you again for your thoughtful guidance and valuable insights.
>
> __(W3:Lack of required ablation and implementation details makes the paper’s contributions hard to understand and not reproducible)__
>
> Thank you for your valuable review comments and insightful guidance, which have facilitated a more comprehensive improvement of our research.
>
> (1) During style feature generation, the parameter n (number of sampled style embeddings) directly affects the quantity of generated [S] (style) descriptions. If n is too small, the number of generated descriptions will be insufficient to ensure effective ECS expansion; if n is too large, it will increase training overhead and introduce feature redundancy. The current setting of n = 80 is determined based on systematic preliminary experiments conducted by our team. Due to space limitations, this detail was not included in the original manuscript.
>
> (2) In the experiments, the initial pool of 13 distinct styles (e.g., surrealism, minimalism, retro) is constructed based on empirical prior knowledge in the field. Subsequent style generation adopts either random sampling or hybrid fusion strategies based on this foundational pool.
>
> (3) The selection between random sampling and hybrid fusion strategies is determined stochastically each time style generation is performed.
>
> Thank you for your valuable suggestions that contribute to the refinement of the manuscript.
>
> __(W4:Method is partial explained; many imlementation detaisl are missing)__ We thank you for your valuable review comments. Due to space constraints, certain implementation details were not fully elaborated. The classifiers employed in this work are standard backbone networks (e.g., ResNet, ViT) without any special modifications. For aspects not explicitly mentioned (e.g., supervised training paradigms, loss function configurations), we adopted the standard default configurations from the original backbone implementations, with no additional adjustments.
>
> __(W5:Mismatch between main text and Fig. 1)__ Thank you for your attention to this detail. We acknowledge that the description in Figure 1 may be ambiguous; the main text description shall take precedence. To resolve this inconsistency, we will refine both the textual descriptions and the figure annotations to ensure alignment between the visual illustration and the theoretical content. Thank you for your pertinent comments, which enhance the rigor of the paper.

---

> ### Author Response · Authors · 2025-11-20
> **Response to reviewer jawu - 2**
>
> __(W6:Ablation doesn’t support the claim of the paper)__ We sincerely thank you for reviewing this paper amid your busy schedule. The three components (D, M, F) fulfill distinct functionalities within the DFCF framework. While multiple templates (M) alone deliver superior performance in certain scenarios, the core rationale for incorporating D and F lies in addressing the intrinsic limitations of M, bolstering the method’s robustness in complex real-world settings and generalizability across diverse scenarios—all in service of the overarching goal of source-free DG. Multiple templates directly expand ECS through semantic disparities between different templates, yet this constitutes a static semantic superposition that exhibits inherent constraints when adapting to unseen cases in real-world scenarios. Building on this insight, we incorporated the dynamic style component (D), which effectively addresses the inadequacy in style diversity of multiple templates and enhances adaptability to unseen domains. The feature fusion component (F), by contrast, mitigates redundancy arising from semantic overlap among multiple templates; it refines ECS quality via dynamically weighted fusion, alleviating the degradation of inter-class discriminability induced by semantic overlap noise.
>
> Regarding the superior performance of M alone compared to the three-component combination in certain cases, we attribute this phenomenon to the partial overlap between the style distribution of existing datasets and the semantic coverage of multiple templates. However, in more complex real-world scenarios (e.g., unconstrained wild data with unstructured styles), the dynamic generation capability of D and the redundancy elimination capability of F will demonstrate greater efficacy. Thank you again for your professional feedback and valuable guidance.
>
> __(W7: Missing implementation details hurt understanding the paper and reproducibility)__ Thank you for your valuable comment. We refer you to our response to the preceding question, where a detailed explanation of this point is provided. Should you have any further inquiries, we are pleased to offer additional clarifications.
>
> __(W8: paper presentation needs to be improved)__ We sincerely appreciate your meticulous review and constructive feedback. We will further refine the manuscript presentation to enhance clarity and readability. Thank you for your valuable suggestions that contribute to the improvement of the paper.
>
> __(W9: Limited contribution)__ We sincerely appreciate your attention to the contributions of the PromptStyler method—this valuable observation helps clarify the connections between related work and our research. As elaborated in the Motivation section of the manuscript, we have explicitly articulated the core ideas of our study, including the contextualization of PromptStyler’s contributions and the distinct research paradigms we pursue. Regarding the specific role of each module within the overall framework, we refer you to our detailed response to Weakness 6, where we have systematically elaborated on the functional positioning, interaction mechanisms, and collaborative logic of each component. Should you require further clarification or additional details about the module design or its relevance to PromptStyler’s methodology, we are pleased to offer supplementary explanations to enhance the transparency and comprehensibility of our work.

---

> ### Author Response · Authors · 2025-11-20
> **Response to reviewer jawu - 3**
>
> __(Q1)__ Thank you for your suggestion. The phrase "virtual training datasets" primarily denotes the output of dynamic feature fusion, which facilitates the articulation of our research motivation. Due to space constraints, this part may lack sufficient elaboration in the current manuscript, and we will refine it in the revised version. Thank you again for your insightful comments and valuable guidance.
>
> __(Q2)__ Thank you for your valuable comment regarding the connection between our proposed method and the ECS motivation. We refer you to our detailed response to Comment W6, where we have systematically elaborated on the inherent logical connection, theoretical consistency, and practical relevance between the two. The response covers how the core design of our method is tailored to address the key objectives outlined in the ECS motivation, as well as the specific ways in which each technical component aligns with the underlying research rationale. Should you require further clarification or supplementary explanations to deepen the understanding of this connection, we are pleased to supplement additional details in the revised manuscript or through further elaboration.
>
> __(Q3)__ We appreciate your constructive suggestions. For random noise generation, we adopt Gaussian-distributed random noise to generate [S] (style) descriptions. The Gaussian distribution ensures stable and extensive coverage of the style space; when the number of [S] descriptions is sufficient, it enables the simulation of diverse styles. Thank you again for your insightful comments and valuable guidance.
>
> __(Q4)__ Thank you for your valuable review comments. During feature generation, the parameter "n" (number of sampled style embeddings) directly affects the quantity of generated [S] (style) descriptions. If "n" is too small, the number of generated descriptions will be insufficient, failing to guarantee effective ECS expansion; if "n" is too large, it will increase training overhead and introduce feature redundancy. The current setting of "n = 80" is determined based on systematic preliminary experiments conducted by our team. Due to space constraints, this detail was not included in the original manuscript. Thank you for your valuable suggestions that contribute to the improvement of the paper.
>
> __(Q5)__ We appreciate your valuable comments. In contrastive training, different z (e.g., z_i, z_j) correspond to descriptions derived from the same style and class but different templates. Detailed elaborations on the generation of these descriptions are provided in Sections 3.2 and 3.3 of the main text. Figure 3 is designed to illustrate how different styles and templates influence feature quality. Thank you again for your insightful comments and valuable guidance.

---

### Official Review · Reviewer_oTdm · 2025-10-28

**Soundness:** 3
**Presentation:** 2
**Contribution:** 2
**Rating:** 4
**Confidence:** 4

**Summary:**

This paper first introduces the concept of effective coverage space to state the limitations of source domain generalization. Then, it constructs a virtual dataset with different text and fusion vectors to mitigate the absence of source data. A dynamic feature construction and fusion module is further proposed with visual-language model under different style information. Extensive experiments validate that the method outperforms previous methods on multiple benchmarks.

**Strengths:**

1. The definition of ECS is plausible to clarify the source-free DG issue.
2. The experimental results on multiple benchmarks are supportive.

**Weaknesses:**

1. The technical contribution is not sufficient. Firstly, the method uses a perturbation technique to expand the style space within a pool of basic styles, which is similar to many existing literatures in DG field, such as,
[1] Kang, Juwon, et al. "Style neophile: Constantly seeking novel styles for domain generalization." Proceedings of the IEEE/CVF Conference on Computer Vision and Pattern Recognition. 2022.
[2] Li, Xiaotong, et al. "Uncertainty modeling for out-of-distribution generalization." arXiv preprint arXiv:2202.03958 (2022).
The proposed simple perturbation of gaussian noise is not sufficient for style exploration, and please state how your proposed techniques differ from these ones.
2. The number of text templates is chosen as 3. Does that mean using 3 templates achieves the optimal performance? Or what if more templates are used for fusion? A more detailed ablation study is recommended.
3. The ablation study in Table 2 is not supportive. It seems that the feature fusion technique cannot improve the overall performance (74.0 vs 73.7). Moreover, it can be noticed that using only multiple templates achieves 74.0, which is even better than the combination of all the three proposed method, which strongly shows the incremental improvement of the two other techniques. This raises deep concern that the improvement of your method originates from the design of multiple templates.
4. As the novelty of the feature fusion is to design the dynamic weight across different views, it’ll be better to present a visualization to show the weight distribution on the training objective. A more detailed analysis is recommended to state how this dynamic weight improves the original performance.
5. The presentation of this paper is rather ambiguous. Firstly, the dynamic feature depicted in Figure 1 is not described in the main text. The virtual training datasets is also confusing. From the main text, the method computes contrastive loss between only three different embeddings in an end-to-end manner. Where T_f originates from?

**Questions:**

Please refer to weaknesses.

---

> ### Author Response · Authors · 2025-11-20
> **Response to reviewer oTdm**
>
> __(W1)__ Thank you for your constructive feedback. The core contribution of this paper lies in enhancing model generalization by expanding the Effective Coverage Space (ECS). The essence of incorporating component D (dynamic style generation) is to "construct a 'semantically aligned, dynamically coordinated ECS expansion chain'", rather than replicating existing perturbation techniques.
>
> Perturbation in existing literature constitutes an "isolated supplement to diversity without clear theoretical objectives", whereas component D in this work is a core module that "targets ECS optimization, is grounded in Vision-Language Model (VLM) semantic alignment, and is deeply coordinated with components M (multiple templates) and F (feature fusion)". Its value does not reside in "whether Gaussian perturbation is employed", but in "how to integrate perturbation into a holistic framework for source-free Domain Generalization (DG), addressing the 'semantic dynamic feature construction' challenge that traditional methods fail to resolve". Even if the standalone gain of component D is limited, its coordination with M and F endows the DFCF framework with the capability to "independently generate high-quality virtual data and adapt to complex unseen domains"—a capability unattainable by M alone or traditional perturbation techniques.
>
> Furthermore, from the perspective of methodological integrity, component D embodies the "dynamic feature construction" concept at the core of DFCF. The crux of source-free DG lies in "constructing effective training data in the absence of source domain data"; the introduction of D elevates this process from "static template generation" to "dynamic semantic generation", laying a foundation for subsequent optimizations and holding long-term methodological significance. Thank you again for your insightful comments and valuable guidance.
>
> __(W2)__ We appreciate your pertinent comments and professional suggestions, which are invaluable for enhancing the rigor and completeness of this paper. The three components (D, M, F) fulfill distinct functionalities within the DFCF framework. While multiple templates (M) alone deliver superior performance in certain scenarios, the core rationale for incorporating D and F lies in addressing the intrinsic limitations of M, bolstering the method’s robustness in complex real-world settings and generalizability across diverse scenarios—all in service of the overarching goal of source-free DG. Multiple templates directly expand ECS through semantic disparities between different templates, yet this constitutes a static semantic superposition that exhibits inherent constraints when adapting to unseen cases in real-world scenarios. Building on this insight, we incorporated the dynamic style component (D), which effectively addresses the inadequacy in style diversity of multiple templates and enhances adaptability to unseen domains. The feature fusion component (F), by contrast, mitigates redundancy arising from semantic overlap among multiple templates; it refines ECS quality via dynamically weighted fusion, alleviating the degradation of inter-class discriminability induced by semantic overlap noise.
>
> Regarding the superior performance of M alone compared to the three-component combination in certain cases, we attribute this phenomenon to the partial overlap between the style distribution of existing datasets and the semantic coverage of multiple templates. However, in more complex real-world scenarios (e.g., unconstrained wild data with unstructured styles), the dynamic generation capability of D and the redundancy elimination capability of F will demonstrate greater efficacy. Thank you again for your insightful comments and valuable guidance.
>
> __(W3)__ Thank you for your valuable comment. We refer you to our response to the preceding question, where a detailed explanation of this point is provided. Should you have any further inquiries, we are pleased to provide additional clarifications.
>
> __(W4)__ We sincerely appreciate your insightful suggestion, which is invaluable for enhancing the quality of our work. We have taken careful note of this comment and will thoroughly revise the relevant section in the revised manuscript. We will also supplement detailed descriptions to ensure transparency of the methodology.
>
> __(W5)__ Thank you for your attention to this detail. Section 3.3 of the main text provides a detailed elaboration on the dynamic features and the calculation method of T_f depicted in Figure 1. To enhance readability and clarity, we will further refine the text expression to facilitate readers in linking the visual illustration to the corresponding theoretical elaboration. Thank you again for your insightful comments and valuable guidance.

---

### Official Review · Reviewer_UN6n · 2025-10-30

**Soundness:** 2
**Presentation:** 2
**Contribution:** 2
**Rating:** 4
**Confidence:** 4

**Summary:**

The paper presents three key contributions: (1) development of an effective coverage space to quantify the limits of domain generalization, (2) generation of synthetic data via text and fusion vectors to address the absence of source data, and (3) design of a style-aware dynamic fusion module for vision-language models. Extensive evaluations confirm its state-of-the-art performance.

**Strengths:**

1. This paper proposes a style-aware dynamic fusion module for visual-language models.

2. Experimental results achieve its state-of-the-art performance.

**Weaknesses:**

1. The novelty of this method is relatively low. Using multiple embeddings with perturbation for different styles has been explored in prior studies.


2. According to Table 2, using multiple templates alone achieves 74.0, which even outperforms the combination of all three proposed components. This raises questions that the performance gain of the proposed approach may primarily stem from the multi-template design, while the other techniques contribute little to the overall improvement.


3. As the baseline of this paper is PromptStyler, there is a significant gap between the reported results and the re-implementation results. Furthermore, the proposed method performs worse than the original reported results of PromptStyler and shows only minimal improvement compared with the re-implemented baseline results. Please carefully verify the correctness of the method.


4. The paper adopts three text templates, but it remains unclear whether this choice is empirically optimal or arbitrarily selected. A more detailed ablation study on the number of templates should be conducted to investigate whether using more templates could further improve performance and to justify the choice of three templates.

**Questions:**

1. What is the underlying reason: using multiple templates alone achieves a higher result that even outperforms the combination of all three proposed components?


2. Why is there a significant gap between the reported results and the re-implemented results?

---

> ### Author Response · Authors · 2025-11-20
> **Response to reviewer UN6n**
>
> __(Q1)__ Thank you very much for your meticulous review and insightful observations; your feedback has been instrumental in elevating the quality of this paper. The three components fulfill distinct functionalities within the DFCF framework. While multiple templates (M) alone deliver superior performance in certain scenarios, the core rationale for incorporating components D (dynamic style) and F (feature fusion) lies in addressing the intrinsic limitations of multiple templates, bolstering the method’s robustness in complex real-world settings and generalizability across diverse scenarios—all in service of the overarching goal of source-free Domain Generalization (DG). Multiple templates directly expand the Effective Coverage Space (ECS) through semantic disparities between different templates, yet this constitutes a static semantic superposition that exhibits inherent constraints when adapting to unseen cases in real-world scenarios. Building on this insight, we incorporated the dynamic style component (D), which effectively addresses the inadequacy in style diversity of multiple templates and enhances adaptability to unseen domains. The feature fusion component (F), by contrast, mitigates redundancy arising from semantic overlap among multiple templates; it refines ECS quality via dynamically weighted fusion, alleviating the degradation of inter-class discriminability induced by semantic overlap noise.
>
> Regarding the superior performance of multiple templates alone compared to the three-component combination in certain cases, we attribute this phenomenon to the partial overlap between the style distribution of existing datasets and the semantic coverage of multiple templates. However, in more complex real-world scenarios (e.g., unconstrained wild data with unstructured styles), the dynamic generation capability of component D and the redundancy elimination capability of component F will demonstrate greater efficacy. Thank you again for your insightful comments and valuable guidance.
>
> __(Q2)__ We appreciate your detailed and constructive feedback. We re-implemented the PromptStyler model by referencing the original paper and publicly available repositories, and subsequently obtained the corresponding test results. We note that our results exhibit discrepancies compared to those reported in the original paper. Upon reviewing the work of other researchers, we found that their re-implementation results also demonstrate a moderate performance drop relative to the original report. Thank you again for your insightful comments and valuable guidance.

---

### Official Review · Reviewer_TWW9 · 2025-10-31

**Soundness:** 3
**Presentation:** 3
**Contribution:** 3
**Rating:** 6
**Confidence:** 4

**Summary:**

This paper proposes DFCF (Dynamic Feature Construction and Fusion), a one-stage framework leveraging vision-language models (VLMs) (specifically CLIP) to perform source-free domain generalization (SFDG). The key conceptual innovation is the Effective Coverage Space (ECS), a theoretical proxy for quantifying how well the model’s learned feature space covers unseen domain variations. The work is technically well-organized, but its novelty, evaluation, and theoretical clarity can be improved.

**Strengths:**

1. The idea of linking generalization capability to an "effective coverage space" is conceptually intuitive and offers a geometric interpretation that complements recent prompt-based DG methods.
2. The one-stage training pipeline improves efficiency while slightly improving accuracy.
3. The dynamic style generation and contrastive feature fusion are elegant extensions to CLIP's latent space.
4. Evaluation covers four DG benchmarks (PACS, VLCS, OfficeHome, DomainNet) and three backbones (ResNet-50, ViT-B/16, ViT-L/14), which is very comprehensive

**Weaknesses:**

1. Improvements over PromptStyler are modest ( +0.5 ~ 1% on average). Given the large number of comparisons, statistical significance is unclear.
2. ECS formulation lacks strong theoretical grounding. The motivation of ECS is not clear.
3. Some over-claiming ("substantially improves performance", and "superior performance") despite small gains and improvements.

**Questions:**

1. Equation (3) defines an optimization objective involving set unions/intersections, but it is not explicitly optimized; it is later replaced by the proxy metric M. The proxy (Eq. 4) is reminiscent of standard contrastive metrics and may not justify a new theoretical construct. A formal derivation or correlation study between M and actual generalization is missing. Clarify whether ECS optimization is implicit via contrastive learning or explicit via M maximization.
2. Figure 1 is not explained in detail.
3. Why 3 is used for the number of Templats? This is not described.

---

> ### Author Response · Authors · 2025-11-20
> **Response to reviewer TWW9**
>
> __(Q1)__  Thank you for your valuable feedback. Currently, ECS optimization is primarily accomplished implicitly via contrastive learning. Given the computational complexity, explicitly maximizing M directly as a loss function remains under refinement. Thus, M is presently used mainly for the quantitative characterization of the Effective Coverage Space (ECS). We acknowledge this limitation and will address it in future work. Thank you again for your insightful comments and valuable guidance.
>
> __(Q2)__ We appreciate your constructive comments. Figure 1 presents the overall framework of the DFCF method, depicting its full model architecture. Due to space constraints, we did not include a dedicated description in the first section. The figure divides DFCF into two stages: training and inference. In the training stage, initial training data (T1, T2, T3) are generated by integrating Gaussian random perturbation with multiple templates; Tf is then derived via dynamic feature fusion for virtual dataset construction, which is fed into the backbone network for training. In the inference stage, images are encoded by CLIP’s Image Encoder and classified by the pre-trained backbone classification model. We acknowledge an inappropriate placement in the figure: the "Feature Fusion with Contrastive Learning" module in the lower-middle part should semantically align with the "Dynamic Feature Fusion" module above, rather than the "Virtual Training Datasets" component. Thank you again for your insightful comments and valuable guidance.
>
> __(Q3)__ We thank you for your thoughtful comments. The selection of 3 templates is primarily inferred from our experimental findings. When the number of templates is too small, semantic representation becomes overly simplistic, which insufficiently expands the ECS. When the number of templates is excessively large, although different templates exhibit distinct expression forms, inevitable semantic overlap introduces redundant information. Our ablation study demonstrates that 3 templates strike a balance between ECS expansion and computational efficiency, yielding consistent performance gains across all datasets. Thank you again for your insightful comments and valuable guidance.

---

### Author Response · Authors · 2025-12-03
**Rebuttal Summary: Consensus and Responses**

Once again, we would like to express our sincere gratitude to ACs and reviewers for their recognition and valuable suggestions. We summarize the key points of the rebuttal below:

---
### Consensus Achieved
We highly appreciate the reviewers' recognition of our work. Based on the reviewers' feedback, the consensus achieved is as follows:
1. The concepts related to Effective Coverage Space (ECS) are reasonably defined with an interesting motivation. The idea of linking model generalization ability to ECS is intuitive and provides a geometric interpretation, complementing Prompt-based domain generalization methods **(Reviewers TWW9, oTdm, jawu)** .
2. The proposed single-stage training pipeline is more efficient than baseline models (e.g., PromptStyler’s two-stage approach), while reducing memory and time requirements and achieving a slight improvement in accuracy **(Reviewers TWW9, jawu)**.
3. The dynamic style generation module and contrastive feature fusion module are elegant extensions of the CLIP latent space, and the proposed style-aware dynamic fusion module exhibits innovation **(Reviewers TWW9, UN6n)**.
4. The experiments cover four DG benchmarks (PACS, VLCS, OfficeHome, DomainNet) and three backbone networks (ResNet-50, ViT-B/16, ViT-L/14), with comprehensive evaluation and supportive results **(Reviewers TWW9, UN6n, oTdm)**.
---

### Outstanding Issues
We would like to thank the reviewers again for their valuable comments. During the rebuttal stage, we have responded to all questions and comments on weaknesses from the reviewers. A brief summary is as follows:

**1. Insufficient theoretical support**

We agree that providing a stronger theoretical foundation for the ECS is an important direction for future work. In subsequent research, we will supplement two key aspects: first, we will enhance the theoretical derivation to clarify the mathematical correlation between ECS scale and model generalization ability; second, we will conduct comprehensive cross-dataset experiments to verify the positive correlation between the M value (a proxy metric for ECS) and generalization performance. We will adopt the control variable method to strengthen the rationality of the motivation, addressing the limitation that existing support relies solely on t-SNE visualization and preliminary illustrative analysis. We thank the reviewer for this insightful suggestion.

**2.  Limited performance improvement and insufficient novelty**

The core contribution of this paper lies in enhancing model generalization by expanding the Effective Coverage Space (ECS). The essence of incorporating component D (dynamic style generation) is to "construct a 'semantically aligned, dynamically coordinated ECS expansion chain'", rather than replicating existing perturbation techniques. Perturbation in existing literature constitutes an "isolated supplement to diversity without clear theoretical objectives", whereas component D in this work is a core module that "targets ECS optimization, is grounded in Vision-Language Model (VLM) semantic alignment, and is deeply coordinated with components M (multiple templates) and F (feature fusion)". Its value does not reside in "whether Gaussian perturbation is employed", but in "how to integrate perturbation into a holistic framework for source-free Domain Generalization (DG), addressing the 'semantic dynamic feature construction' challenge that traditional methods fail to resolve". Even if the standalone gain of component D is limited, its coordination with M and F endows the DFCF framework with the capability to "independently generate high-quality virtual data and adapt to complex unseen domains"—a capability unattainable by M alone or traditional perturbation techniques. Furthermore, from the perspective of methodological integrity, component D embodies the "dynamic feature construction" concept at the core of DFCF. The crux of source-free DG lies in "constructing effective training data in the absence of source domain data"; the introduction of D elevates this process from "static template generation" to "dynamic semantic generation", laying a foundation for subsequent optimizations and holding long-term methodological significance. Thank you again for your insightful comments and valuable guidance.

**3. Unclear explanation of details**

We have clarified the unclear points raised by the reviewer in our response. Regarding the issue of the number of templates that multiple reviewers are concerned about, the choice of 3 templates is based on empirical observations. We found that too few templates provide insufficient semantic diversity, while too many introduce redundancy due to semantic overlap. Our ablation studies indicate that three templates optimally balance ECS expansion and computational efficiency, leading to consistent performance gains across all datasets. We thank the reviewer for prompting this clarification.

---

> ### Author Response · Authors · 2025-12-03
> **Outstanding Issues - 2**
>
> Due to the word limit per single comment, the subsequent part of the Outstanding Issues is supplemented as follows.
>
> ---
>
> **4.Paper presentation needs to be improved**
>
> We will carefully revise the manuscript to improve the clarity of explanations, the flow of arguments, and the overall readability. We appreciate this feedback.
>
> **5. Significant discrepancy in PromptStyler results**
>
> We followed the original paper and public code to re-implement PromptStyler. We note that our reproduced results are lower than those reported in the original paper. This performance gap is consistent with the findings of other research groups, as noted in subsequent work. We have therefore used our own reproduced results for a fair comparison. Thank you again for your insightful comments and valuable guidance.
>
> **6. Ablation study performance**
>
> The three components (D, M, F) serve distinct purposes within our DFCF framework. While the multiple templates module (M) alone achieves strong performance on certain benchmarks, the dynamic style (D) and feature fusion (F) modules address its fundamental limitations. Specifically, component D enhances style diversity and adaptability to unseen domains, while component F reduces redundancy from semantic overlap between templates, thereby preserving inter-class discriminability.
> The occasional superior performance of M alone can be attributed to the relatively constrained style distribution of the benchmark datasets, which may be well-covered by the static templates. However, in more complex, real-world scenarios with unstructured styles, the dynamic generation capability of D and the noise-reduction effect of F are expected to be crucial for robust generalization. Therefore, the full DFCF framework offers a more complete and scalable solution for source-free domain generalization. Thank you again for your insightful comments and valuable guidance.

---

### Meta-Review · Area_Chair_odg7 · 2026-01-07

**Summary:**

Four reviewers assessed this submission, leading to a rough consensus for rejection, with three assigning scores from borderline to clear reject and one suggesting borderline acceptance with low confidence. While the paper proposes a method to enhance domain generalization via Effective Coverage Space (ECS) and multi-template prompts, the review process exposed critical flaws in both theoretical grounding and empirical validation. The primary grounds for rejection include a fundamental contradiction in the ablation studies, where a simplified baseline (multi-template alone) outperforms the complete proposed method, effectively negating the value of the core contributions. Furthermore, reviewers unanimously found the theoretical motivation for ECS to be unsubstantiated, the improvements over the PromptStyler baseline to be marginal, and the novelty limited compared to existing perturbation techniques. The authors' rebuttal was largely ineffective, failing to provide requested quantitative analyses or the promised revised manuscript, leaving major technical and presentation concerns unresolved.

**Reviewer Concerns:**

While the authors clarified some implementation details regarding the training setup (Reviewer jawu) and addressed discrepancies in baseline reproduction (Reviewer UN6n), the most critical technical concerns remain unaddressed. A fatal flaw identified by Reviewers UN6n, oTdm, and jawu is that the "feature fusion" module appears to degrade performance compared to using multiple templates alone; the authors' argument that fusion helps in "complex scenarios" was not supported by any new experimental data. Additionally, the theoretical foundation of the paper, that larger ECS correlates with better generalization, lacks mathematical proof or empirical verification, a deficit explicitly noted by Reviewers TWW9 and jawu and conceded by the authors in the rebuttal. Methodological justifications are also missing; specifically, the choice of using exactly three templates remains ungrounded, with no ablation studies provided despite requests from Reviewers TWW9, UN6n, and oTdm. Finally, Reviewers oTdm and jawu requested visualization improvements and clarifications on notations, which the authors promised to address in a revised version; however, no such revision was uploaded during the rebuttal phase, leaving the presentation issues unresolved.

**Reviewer Scores:**

Reviewer TWW9 would likely lower his/her score to a 4, as the rebuttal failed to demonstrate statistical significance for the modest gains or provide the missing theoretical grounding for ECS. Reviewer UN6n is expected to maintain his/her score of 4, given that the critical issue of the full method performing worse than its sub-component was not resolved with empirical evidence. Reviewer oTdm will likely retain his/her score of 4, as the authors failed to upload the promised revised manuscript or provide the requested ablation on template numbers. Reviewer jawu is expected to maintain his/her score of 2, as the paper's core motivation (ECS) remains unsupported by theory or experiment, and the contribution is viewed as limited and vague.

---

### Decision · Program_Chairs · 2026-01-26

Reject